# Circular RNA ITCH: An Emerging Multifunctional Regulator

**DOI:** 10.3390/biom12030359

**Published:** 2022-02-24

**Authors:** Kaixin Su, Qiao Yi, Xiaohan Dai, Ousheng Liu

**Affiliations:** 1Academician Workstation for Oral-Maxilofacial and Regenerative Medicine, Central South University, Changsha 410008, China; sukaixin@csu.edu.cn (K.S.); 9320189yiq@csu.edu.cn (Q.Y.); 2Hunan Key Laboratory of Oral Health Research, Central South University, Changsha 410008, China; 3Hunan 3D Printing Engineering Research Center of Oral Care, Central South University, Changsha 410008, China; 4Hunan Clinical Research Center of Oral Major Diseases and Oral Health, Central South University, Changsha 410008, China; 5Xiangya Stomatological Hospital, Central South University, Changsha 410008, China; 6Xiangya School of Stomatology, Central South University, Changsha 410008, China

**Keywords:** circular RNAs, circ-ITCH, ITCH, biological functions, pathological mechanisms

## Abstract

In the last decade, numerous circRNAs were discovered by virtue of the RNA-Seq technique. With the deepening of experimental research, circRNAs have brought to light the key biological functions and progression of human diseases. CircRNA ITCH has been demonstrated to be a tumor suppressor in numerous cancers, and recently it was found to play an important role in bone diseases, diabetes mellitus, and cardiovascular diseases. However, the functions of circ-ITCH have not been completely understood. In this review, we comprehensively provide a conceptual framework to elucidate circ-ITCH biological functions of cell proliferation, apoptosis and differentiation, and the pathological mechanisms of inflammation, drug resistance/toxicity, and tumorigenesis. Finally, we summarize its clinical applications in various diseases. This research aimed at clarifying the role of circ-ITCH, which could be a promising therapeutic target.

## 1. Background

In the past years, the human genome has been estimated by new technologies, and most of them were transcribed into RNAs. However, only 1–2% of the human transcriptome encodes for proteins [1]. RNAs without coding potential are named non-coding RNAs (ncRNAs). ncRNAs were discovered, including microRNAs (miRNAs), circular RNAs (circRNAs), and long non-coding RNAs (lncRNAs). These ncRNAs were once considered “transcriptional noise” and not taken seriously. Until recent years, increasing evidence has suggested that ncRNA plays key roles in diverse molecular mechanisms [2,3].

The circRNAs were discovered in 1976 after more than 30 years of being overlooked, and they have gradually become a new molecule of interest nearly a decade later [4,5]. The circRNAs are a class of non-coding RNA molecules that are short of a 5′ cap and a 3′ poly (A) tail; thus, they form a ring structure with covalent bonds [6]. Studies have revealed that circRNA is an abundant and conservative type of RNA that is widely expressed in disease-related tissues, cells and normal tissues [7].

## 2. Introduction

### Circular RNAs(circRNAs): Subtypes and Mechanisms 

Based on molecular origination, circRNAs can be divided into separate subtypes: exonic circRNAs (ecRNA), intronic circRNAs (ciRNA) and exon-intron circRNAs (ELciRNA) [8]. Some of the ciRNAs are isolated in the nucleus, while the ecRNA is exported to the cytoplasm and can be widely detected within it. Currently, ecRNA accounts for the vast majority of circRNA and is the most studied [9]. Moreover, as the product of variable splicing, the circRNAs proposed biological functions are as follows:

**(a) Acting as a miRNA sponge.** circRNAs function as competing endogenous RNAs (ceRNAs), which bind miRNAs and therefore inhibit the target genes. This is the most critical function of circRNA compared to other functions. The most classic research has been conducted upon circRNA ciRS-7, and it has been found to have more than 70 binding sites for miR-7; thus, it can strongly suppress the expression of miR-7 target gene CDR1 [10] (Figure 1a).

**(b) Regulating gene transcription.** Although most circRNAs function in the cytoplasm, ciRNAs that remain in the nucleus also have a significant role, interacting with RNA polymerase II (Pol II) machinery and modulating host transcription. ElciRNAs promote parent gene transcription by binding to the small nuclear ribonucleoprotein U1 (snRNP U1) to form a complex and further interact with Pol II [11]. For instance, circPABPN1 suppressed PABPN1 translation through extensively binding to HuR protein, which effects PABPN1 mRNA in HeLa cells [12] (Figure 1b).

**(c) Interacting with proteins.** circRNAs can also act as protein scaffolds by interacting with RNA-binding proteins (RBP). For example, circ-Foxo3 can bind with p21 and CDK2 to form ternary complexes, thereby repressing cell cycle progression at the G1 and S phase [13] (Figure 1c).

**(d) Encoding function.** Recent studies have shown that some ecRNAs can be effectively translated into proteins or peptides in a cap-independent manner. Both internal ribosome entry site (IRES)- and 5′ UTR N^6^-methyladenosine (m^6^A)-mediated translation initiation are considered potential translation mechanisms of circRNAs in cytoplasm [14,15]. In 2018, Yang et al. reported that circFBXW7 could encode protein FBXW7-185aa, which inhibits the proliferation and migration of glioma cancer cells by regulating the stability of c-Myc [16] (Figure 1d).

Recently, a great deal of evidence has revealed that circRNAs are widely expressed in a disease-specific manner. Over the past few years, the application of circRNAs have been universally acknowledged in tumorigenesis and cancer progression [9]. The role of circRNAs in solid tumors are targetable markers such as oncogenes and tumor suppressors [7]. As research progresses, the functions of circRNAs in other diseases, such as cardiovascular diseases, age-related diseases, viral infections, and neurological disorders are emerging [17,18,19,20,21].

## 3. The Formation of Circ RNA ITCH

In 1998, the itchy E3 ubiquitin protein ligase (ITCH) function of regulating protein stability was first discovered in non-agouti-lethal 18H mice [22]. As a member of the E3 ubiquitin ligases, the ITCH protein is implicated in biological functions such as immunity, tumorigenesis, and inflammatory responses [23,24]. ITCH prevents the development of autoimmune diseases by regulating immune cell biology and controlling cancer progression by targeting p63, p73, Notch1, and Dvl2 [23,24,25]. 

Circular RNA itchy E3 ubiquitin protein ligase (circ-ITCH, circBase ID: hsa_circ_0001141) was first reported by Memczak et al. [5]. Circ-ITCH is generated from the several exons of ITCH located on chromosome 20q11.22 [26] (Figure 2). The expression correlation between circ-ITCH and its parent gene ITCH has been confirmed [27]. Because circ-ITCH shares the same miRNA binding sites with the 3′-untranslated region (UTR) of ITCH [5], circ-ITCH might regulate ITCH gene transcription by acting as a miRNA sponge for miR-7, miR-17, and miR-214 [28].

## 4. The Biological and Pathological Functions of Circ RNA ITCH

In the past ten years, circ-ITCH has been demonstrated to play a crucial role in the progression of malignant cancers. Moreover, evidence that circ-ITCH contributes to non-neoplastic disease has gradually emerged. According to the existing research on circ-ITCH, we summarize biological and pathological processes that hope to provide vital clues about disease in the future.

### 4.1. Circ-ITCH and Cell Proliferation

Circ-ITCH effects cell proliferation mainly through regulating target miRNAs and thus indirectly effecting the parental gene, ITCH. The Wnt signaling pathway is highly conserved in species’ evolution that plays a vital role in the development of cell proliferation, apoptosis, tissue regeneration, and other physiological processes [29]. Dishevelled (Dvl) is an effector molecule that stabilizes β-Catenin in the free state of the cytoplasm by inhibiting APC/Axin/GSK3β proteolytic complex formation [30]. Subsequently, β-Catenin enters the nucleus and binds to TCF transcription factors to trigger the transcription of Wnt target genes. Thus, the ITCH proteins promote the degradation of Dvl and consequently inhibit canonical Wnt signaling [31]. The circ-ITCH acts as a sponge and absorbs some miRNAs, such as miR-214, miR-7, miR-20a, and miR-17, which bind to the 3’-UTR of ITCH, thereby regulating ITCH expression [32]. Taken together, circ-ITCH inhibits the Wnt/β-catenin signal activation through the parental gene ITCH. The classical circ-ITCH-miRNA-ITCH-Wnt/β-catenin signaling pathway exerts anti-tumor effects through suppressing cancer cell proliferation (Figure 3) and inducing G1 cell cycle arrest in colorectal cancer, breast cancer, glioma, esophageal squamous cell carcinoma, and many others [27,28,32,33]. In addition to ITCH protein, circ-ITCH can also suppress papillary thyroid cancer development and progression by acting as the sponge of miR-22-3p and regulating the miR-22-3p/CBL/Wnt/β-catenin pathway [34]. Moreover, in prostate cancer, upregulation of circ-ITCH inhibited cell proliferation and induced cell apoptosis through binding miR-197 [35]. Circ-ITCH has been shown to reduce hepatocellular carcinoma cell growth through targeting miR-224-5p and overexpressing its target gene, MafF, which is widely downregulated in multiple cancers [36]. Similarly, circ-ITCH was demonstrated to stimulate SASH1 levels by sponging miR-106a-5p to affect glioma cell proliferation and invasion [37]. Knockdown of circ-ITCH could enhance proliferation and suppress apoptosis by activating miR-382-5p/TOP1 and NF-κB signaling in the LPS-induced MRC-5 cells [38]. Overall, in almost all cancers, circ-ITCH, which is known as a tumor suppressant, inhibits cell proliferation and cancer growth rate.

### 4.2. Circ-ITCH and Cell Apoptosis/Cell Cycle

Circ-ITCH is involved in mediating cell apoptosis and the cell cycle by regulating the expression of apoptosis-related proteins and related signaling pathways. Circ-ITCH curbs myocardial cell apoptosis by decreasing levels of caspase 3, p53 and PARP via the miR-17-5p/Wnt/β-catenin signaling pathway [39]. PTEN protein is a potent tumor inhibitor, which suppresses cell growth and promotes apoptosis in the nucleus and tumor microenvironment [40]. In nasopharyngeal carcinoma, circ-ITCH upregulation induces the expression of PTEN by targeting miR-214, inhibiting the tumorigenesis [41]. Similarly, in bladder cancer cells, circ-ITCH promotes apoptosis and arrests cell cycle at the G1/S phase by the circ-ITCH/miR-17, miR-224/p21, and PTEN axis [42]. Circ-ITCH sponging miR-22 in osteosarcoma effects apoptosis and the cell cycle by restraining PTEN/PI3K/AKT and SP-1 pathways [43]. Based on the above evidence, it can be inferred that the circ-ITCH/miRNA/PTEN/mTOR axis may play a pivotal role in cell apoptosis and growth. Downregulation of circ-ITCH activates some apoptotic proteins, such as bax, cleaved caspase 3 and cleaved caspase 9, and cell cycle-related regulatory factors including CDK2, cyclin E1, p73, and c-Myc in nucleus pulposus (NP) cells, subsequently inducing apoptosis and cell cycle arrest [44]. In oral squamous cell carcinoma (OSCC), circ-ITCH serves as a tumor suppressor by functioning as miR-421 sponge, subsequently increasing expression on PDCD4 [45]. Together, this research indicates that circ-ITCH is closely related to the regulation of cell apoptosis and cell cycle (Table 1).

### 4.3. Circ-ITCH and Cell Differentiation

Recently, circ-ITCH was found to participate in normal cell differentiation, especially within stem cells. Circ-ITCH was identified to differentially express during the osteogenic differentiation of periodontal ligament stem cells (PDLSCs) compared with normal PDLSCs. Further bioinformatic analysis revealed that circ-ITCH was predicted to interact with miR-34a and miR-146a to promote PDLSC osteogenic differentiation via the MAPK pathway [46]. In bone-marrow-derived mesenchymal stem cells (BMSCs), circ-ITCH overexpression promoted osteogenic differentiation by sponging miR-214 in order to upregulate YAP1 expression [47]. Thus, circ-ITCH appears to play a significant role in development and cell differentiation; however, the mechanism is not fully known and requires further study (Table 2). 

### 4.4. Circ-ITCH and Inflammation 

Inflammation is a complex process that is mediated by inflammatory cytokines and different immune cells [48]. Out-of-control inflammation can cause a range of pathological reactions, such as severe infections, sepsis, atherosclerosis, and type 2 diabetes [49]. Acute lung injury (ALI) is an early stage of sepsis, which is a systemic inflammatory response syndrome (SIRS). The study confirmed the positive impacts of the circ-ITCH-mediated miRNA–mRNA axis on LPS-induced MRC-5 cells, which might be a therapeutic target for LPS-induced inflammation in ALI patients [38]. Of note, circ-ITCH has been reported to have an anti-inflammatory effect on human cells and tissue [50]. Circ-ITCH regulates inflammation with emphasis on cytokines and proinflammatory factors such as TNF-α, IL-1β, and IL-6 through mRNA or miRNA. Research has shown that the levels of TNF-α and IL-1β were highly expressed in the serum of the rat intervertebral disc degeneration (IVDD) model; however, these changes can be reversed by circ-ITCH overexpression [44]. In the high-glucose-induced rat mesangial cells (RMCs), circ-ITCH overexpression could alleviate the increased levels of inflammatory factors (IL-6, IL-1β and TNF-α), and circ-ITCH modulates the inflammatory response by sponging miR-33a-5p to modulate SIRT6 expression [51]. SIRT6 acts as a key regulatory protein for the inflammatory response; it not only regulates cytokines horizontally after transcription, but it also elicits macrophage polarization toward an M1 phenotype and promotes M2 macrophage transformation [52,53]. The overexpression of matrix metalloproteinases (MMPs) is positively correlated with inflammatory regulators in the early stages of inflammation, and the degradation of extracellular substrates can further aggravate the infiltration of inflammatory cells, thereby aggravating inflammation [54]. Circ-ITCH restrains inflammation through repressing the expression of MMP-2 and MMP-9 downstream miR-22 in diabetic retinal pigment epithelial cells [55] (Table 3).

### 4.5. Circ-ITCH and Drug Resistance/Toxicity

Widely used chemotherapeutic drugs may produce serious drug resistance and systemic toxicity. Doxorubicin (DXR) is a classical anti-tumor antibiotic with a strong cytotoxic effect. Circ-ITCH has been demonstrated to elevate drug sensitivity and ameliorates drug toxicity in DXR. Zhou et al. found that circ-ITCH inhibits osteosarcoma resistance to DXR via the miR-524/RASSF6 pathway [56]. The DXR side effect of severe cardiotoxicity limits its clinical applications to a certain extent. Han et al. demonstrated that the expression of circ-ITCH was downregulated in cancer patients who suffered from DXR-induced cardiomyopathy compared to those with healthy heart tissues. When DXR-induced human pluripotent stem cell-derived cardiomyocytes were introduced into cardiotoxic mice, circ-ITCH overexpression could prevent cell injury and dysfunction via the miR-330-5p-SIRT6/BIRC5/ATP2A2 axis [57]. It has also been found that circ-ITCH can also overcome chemotherapy bortezomib resistance in multiple myeloma in vitro and in vivo experiments [58]. Castration-resistant prostate cancer (CRPC) patients are resistant to anti-androgen therapy. Circ-ITCH inhibits the progression of PC-3 cells through sponging miR-17, thereby increasing sensitivity to CRPC [59]. Therefore, we believe that the potential role of circ-ITCH with drug resistance and toxicity deserves further research (Table 4).

### 4.6. Circ-ITCH and Tumorigenesis

#### 4.6.1. Invasion and Metastasis Regulation

A retrospective case analysis revealed that 324 prostate cancer patients with downregulated circ-ITCH expression showed a correlation with high lymph node metastasis [60]. Several researchers demonstrated that circ-ITCH participated in the regulation of invasion and metastasis by influencing related protein factors or inhibiting the epithelial-to-mesenchymal transition (EMT). For instance, circ-ITCH affects the EMT pathway and suppresses metastasis of gastric cancer by sponging miR-199a-5p and thereby increasing Klotho expression [61]. Moreover, in osteosarcoma cells, circ-ITCH could promote epidermal growth factor receptor (EGFR) protein levels by activating the EGFR/ERK pathway via miR-7 to promote cancer metastasis, migration, and invasion [62]. Otherwise, circ-ITCH directly binds to miR-93-5p, thus regulating cervical cancer cell metastasis via targeting forkhead box K2 (FOXK2) [63]. Finally, circ-ITCH functions as a ceRNA through activation of the miR-106b-5p/PDCD4 axis to block clear cell renal cell carcinoma growth and suppress metastasis [64].

#### 4.6.2. Cell Metabolism Regulation 

To promote the rapid proliferation of cancer, tumor cells rely on the glycolytic pathway to obtain energy, which is a primitive metabolic pathway, called the Warburg effect. Circ-ITCH has been proven to regulate cellular metabolism-related factors by binding target miRNA. For example, circ-ITCH suppresses glycolysis of ovarian cancer cells by competitively sponging miR-106a to enhance the expression of E-cadherin (CDH1) [65]. In addition, a study revealed that circ-ITCH downregulated the expression of glucose transporter type 1 (GLUT1) and suppressed glucose uptake to inhibit melanoma cell glycolysis and progression [66]. Together, these studies provide new evidence that circ-ITCH is closely related to the regulation of cancer cell metabolism (Table 5).

## 5. The Role of circ-ITCH in Diseases

### 5.1. Diabetes Mellitus

Diabetes mellitus (DM) is a chronic metabolic disease characterized by insufficient insulin secretion from pancreatic β cells and leads to high blood glucose levels [67]. Currently, in 2019, according to the International Diabetes Federation statistics, there are about 463 million diabetes patients worldwide [68]. The complications of DM, including diabetic retinopathy (DR), diabetic nephropathy (DN), and diabetic cardiomyopathy (DCM) can cause multiple organ system damage, thereby endangering the lives of patients. The ability of circ-ITCH to restrain the development of diabetes-related complications is of particular concern. DR is one of the most common microvascular complications of diabetes. The study by Zhou et al. demonstrated that circ-ITCH could reduce an increase in MMP-2, MMP-9, and TNF-α in diabetic rats via the inhibition of miR-22. Therefore, circ-ITCH contributes to the restraint of DR progression by preventing the neovascularization and Inflammation [55]. DN is a glomerulopathy, which is mainly caused by vascular damage. In streptozotocin-induced diabetic mice, circ-ITCH overexpression alleviated renal inflammation and fibrosis by the miR-33a-5p/SIRT6 axis [51]. The relationship between diabetes and its complications and circ-ITCH is limited to in vitro cell and animal experiments. In addition, whether circ-ITCH can directly affect β cell proliferation, the glucose intake of β cells, and insulin synthesis and secretion processes deserves further exploration.

### 5.2. Cardiovascular Diseases 

Due to the unhealthy lifestyle and the increasing lifespan of the population, cardiovascular diseases (CVDs) are increasing consistently each year [69]. As a leading cause of morbidity illness, CVD leads to high mortality worldwide and brings heavy social and economic pressure [70]. Of note, the biogenesis and function of numerous circRNAs in the cardiovascular system have been demonstrated [71]. Studies have confirmed that circRNAs are associated with the development of cardiovascular diseases, such as atherosclerosis, myocardial infarction, and heart failure. Zhang et al. demonstrated that circ-ITCH could reverse myocardial ischemia-reperfusion injury (I/R) via the miR-17-5p/Wnt/β-catenin pathway in H2O2-treated H9c2 cells (a rat myocardial cell line), an in vitro model of myocardial I/R [39]. On the other hand, overexpression of circ-ITCH relieves doxorubicin-induced cardiomyocyte injury by acting as a sponge of miR-330-5p, and this suggests that circ-ITCH could reverse cardiotoxicity injury, arrhythmias, and even congestive heart failure caused by doxorubicin [57]. Therefore, circ-ITCH is speculated to turn into a therapeutic target used in cardiovascular diseases.

### 5.3. Bone Diseases

An abundance of circRNAs is expressed in a bone disease-specific manner [72,73]. Presently, circ-ITCH plays a crucial role in the diagnosis and target treatment of bone diseases including intervertebral disc degeneration (IDD), osteoporosis, and osteosarcoma. Osteoporosis is a common metabolic disorder characterized by low bone mineral density and high risk of fracture in the elderly, especially in postmenopausal women [74]. Zhong et al. found that circ-ITCH expression was downregulated in 15 human osteoporotic samples compared with 15 normal tissues. Meanwhile, circ-ITCH could alleviate the symptoms of osteoporosis in OVX mice by promoting osteogenic differentiation via the miR-214/YAP1 axis [47]. IDD is a chronic degenerative disease associated with low back pain in the elderly [75]. The expression of circ-ITCH was decreased in IDD patients, while overexpression of circ-ITCH attenuated IDD in rats [44]. It suggests that circ-ITCH may serve as a critical molecular marker in bone aging or cellular senescence. 

Osteosarcoma (OS) is a malignant bone tumor that often occurs in adolescents under the age of 20. In recent years, circ-ITCH has been identified to regulate the progression of osteosarcoma. Low levels of circ-ITCH were expressed in MG63, U2OS, OS732, and Saos-2 cells of osteosarcoma cell lines, and restrained proliferation and promoted apoptosis by sponging miR-22 in MG63 and Saos-2 cells [43]. On the contrary, Li et al. showed that circ-ITCH was highly expressed in U2OS cells of osteosarcoma cell lines when compared with the osteoblast cell line. Circ-ITCH induced the OS cell migration and invasion through the miR-7/EGFR axis [62]. It is worth mentioning that circ-ITCH shows a promoting effect on osteosarcoma by sponging miR-7 [28]; however, in colorectal cancer, esophageal squamous cell carcinoma, and lung cancer, the biological behavior of circ-ITCH sponging miR-7 mainly inhibits cancer migration, invasion, and proliferation through regulation of the ITCH-Wnt/β-catenin pathway. This difference may be associated with the molecular mechanism and tumor cells. All in all, circ-ITCH has emerged as an important regulator of bone formation, aging, metabolism and homeostasis. 

### 5.4. Cancer Solid Tumor

According to relevant studies, circ-ITCH mainly acts as a tumor suppressor in diverse cancer cell lines. Circ-ITCH effects tumorigenesis through a sponge function, and further involves several signaling pathways, such as the Wnt/β-catenin and PI3K/AKT/mTOR signaling pathways. Next, we will summarize the following cancer mechanism and clinical significance of circ-ITCH.

#### 5.4.1. Prostate Cancer

Prostate cancer (PCa) is one of the most common malignant tumors in the male genital system. There is an urgent need to develop new biomarkers and therapeutic targets with both sensitivity and specificity for the early screening and treatment of PCa [76]. By conducting transcriptome research of 144 PCa patients, an average of 7232 circRNAs were expressed in PCa cells, which related to tumor progression. Furthermore, 11.3% of the circRNAs are essential to cell proliferation [77]. The research shows that circRNAs are important molecules and may regulate the behavior of PCa. Huang et al. concluded that circ-ITCH was downregulated in PCa tissues and was correlated with lower pathological T stage, lower lymph node metastasis risk, and longer overall survival [60]. Wang et al. found that low expression of circ-ITCH was associated with poor overall survival and prompted higher preoperative PSA and higher gleason score. The molecular mechanism of circ-ITCH suppressing PCa is by acting as a sponge to miR-17-5p and reversing the expression of HOXB13 [78]. Moreover, overexpression of circ-ITCH suppresses PCa cell proliferation and promotes cell apoptosis through targeting miR-197 [35]. The regular anti-androgen treatment does not work in advanced prostate cancer patients. This occurs when almost all patients ultimately progress to castration-resistant prostate cancer (CRPC), which greatly shortens the overall survival rate [79]. Sponging miR-17, circ-ITCH had an additive inhibitory action on CRPC tumorigenesis by effecting the Wnt/β-catenin and PI3K/AKT/mTOR pathways [59].

#### 5.4.2. Ovarian Cancer

The mortality of ovarian cancer ranks first among all kinds of gynecological tumors. Due to its high rate of mortality, it is urgent to seek promising treatment to improve overall survival rates [80]. The ability of circ-ITCH to act as an ovarian cancer depressor has been confirmed. Luo et al. studied 77 postoperative epithelial ovarian cancer patients to prove that high levels of circ-ITCH indicate smaller tumor size, decreased FIGO stage, and prolonged overall survival [81]. Sequestering miR-106a by circ-ITCH to upregulate CDH1 expression inhibits invasion and glycolysis of ovarian cancer cells [65]. In vivo experiments show that circ-ITCH suppresses epithelial ovarian cancer cell proliferation and promotes apoptosis through targeting miR-10a-α [82]. Circ-ITCH inhibited the malignant ovarian carcinoma progression via the ceRNA mechanism of sponging miR-145 and overexpressing RASA1 shown in vitro and in vivo experiments [83]. Transcriptome analysis and cell experiments have indicated that circ-ITCH is negatively correlated with lncRNA HLUC, and circ-ITCH overexpression may reduce the proliferation of ovarian carcinoma by downregulating lncRNA HULC [84]. Taken together, the effects of circ-ITCH on ovarian cancer progression are highly variable.

#### 5.4.3. Glioma

More and more studies have shown that circRNAs are abundant in neuronal tissues, which indicates that circRNAs are closely related to nervous system diseases. Glioma is the most common malignant tumor in the central nervous system, accounting for over 80% of all malignant brain tumors [85]. As shown in Chen et al.’s study, both the q-PCR results from 48 glioma patients [37] and 60 primary glioma tissues [86] demonstrated that circ-ITCH expression is downregulated in glioma. Research shows that circ-ITCH exerts a tumor-suppressive function upon glioma cells through the miR-106a-5p/SASH1 axis and miR-214/ITCH-Wnt/β-catenin pathway. Thus, circ-ITCH plays an anti-oncogenic role in glioma. 

#### 5.4.4. Gastric Cancer

circ-ITCH within gastric cancer (GC) is in charge of the anti-cancer function through the ceRNA mechanism. Through the use of in vitro and in vivo tests, circ-ITCH was confirmed to be down-regulated in GC tissues and cell lines and also decreased GC growth by the miR-17/ITCH-Wnt/β-catenin pathway [33]. Moreover, the expression of circ-ITCH in 61 pairs of GC tissue samples and 33 pairs of serum exosomes was knocked down when compared with adjacent normal tissues. According to clinical pathological characteristics, circ-ITCH expression relates to GC invasion depth, and further research shows that circ-ITCH suppresses GC cell line metastasis via regulating the miR-199a-5p/Klotho axis [61]. 

## 6. Perspectives and Conclusions

In this review, we summarize the progress of recent research on the biological roles and pathological mechanisms of circ-ITCH. This review has provided evidence that circ-ITCH plays multiple roles in the development of solid cancer tumors, CVD, DM, and bone-related diseases. As we discussed above, circ-ITCH is an emerging multifunctional regulator and may be a useful biomarker for clinical therapy.

However, the current studies of circ-ITCH are limited. For instance, the understanding of circ-ITCH in tumor types is comprehensive and extensive. However, this understanding remains at the cellular and animal levels and needs to be extended to human levels. There is an urgent need for these data to be translated for clinical application in the near future. Furthermore, based on the positive effects that circ-ITCH has on inflammatory diseases, we speculate that circ-ITCH may have an effect on senescence or fibrosis. In addition, the function of circ-ITCH is limited to the ceRNA mechanism. The online circRNA database shows that several RNA-binding proteins interact with circ-ITCH. These data imply that circ-ITCH has other mechanisms that have yet to be discovered but are necessary for the expansion of circ-ITCH research. Whether or not circ-ITCH can modulate immune responses through the ITCH pathway is another route of investigation that deserves to be explored. Besides, we found that the lack of standardized naming rules for circRNAs greatly increased the workload of collecting information. Thus, it is necessary to standardize the circRNA names. 

## Figures and Tables

**Figure 1 biomolecules-12-00359-f001:**
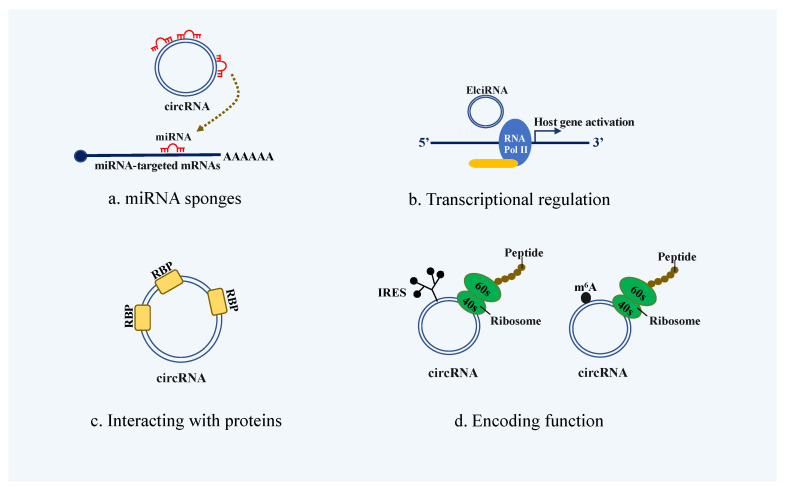
Biological functions of circRNAs. (**a**) circRNAs can function as ceRNA by reducing the expression of mRNAs targeting miRNAs. (**b**) elciRNAs affect the host gene expression by interacting with RNA polymerase II (RNA Pol II) with small nuclear ribonucleoprotein U1 (U1 snRNP) in the nucleus. (**c**) The interactions between circRNAs and RNA binding proteins (RBPs) can modulate these proteins functions. (**d**) Some circRNAs might translate ribosomes in an internal ribosome entry site (IRES)- and 5′ UTR N^6^-methyladenosine (m^6^A)-mediated translation initiation.

**Figure 2 biomolecules-12-00359-f002:**
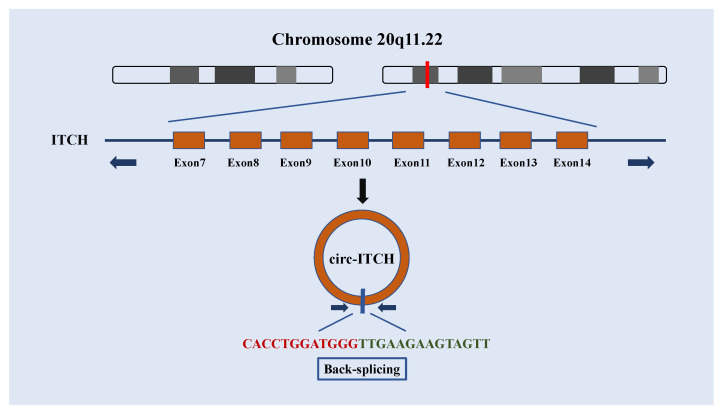
Genomic locus and formation of circ-ITCH. The circ-ITCH located at chromosome 20q11.22. Mature circ-ITCH is derived from exons 7–14 of ITCH gene with 873 bp splice length.

**Figure 3 biomolecules-12-00359-f003:**
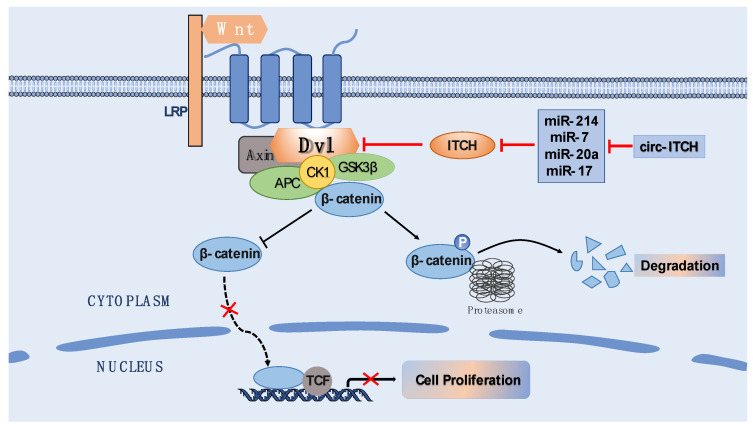
Circ-ITCH-mediated inhibition of Wnt/β-catenin signaling. In the pathway, Dvl protein could remain β-catenin stable by inhibiting APC, Axin and GSK3β to form destruction complexes. The stable accumulation of β-Catenin in cytoplasm enters the nucleus and binds to the TCF transcription factor to initiate transcription of downstream target genes. Instead, the form of destruction complex mediates β-Catenin phosphorylation and, eventually, phosphorylated β-Catenin is degraded by the proteasome. The circ-ITCH acts on miR-214, miR-7, miR-20a and miR-17 through ceRNA mechanism to regulate target parental gene ITCH. The ITCH-mediated degradation of Dvl leads to β-Catenin degradation and inhibition Wnt/β-catenin signaling.

**Table 1 biomolecules-12-00359-t001:** Target and role of circ-ITCH in cell apoptosis/cell cycle.

Disease	miRNA	Related Genes/Pathways	Function Role	Reference
Myocardial ischaemia-reperfusion injury	miR-17-5p	Wnt pathway	Protect myocardial cells from injuries by suppressing apoptosis	[39]
Nasopharyngeal carcinoma	miR-214	PTEN	Inhibit tumor progression	[41]
Bladder cancer	miR-17/miR-224	p21, PTEN	Inhibit tumor progression	[42]
Osteosarcoma	miR-22	PTEN/PI3K/AKT and SP-1 pathways	Suppress proliferation, migration and facilitate apoptosis	[43]
Oral squamous cell carcinoma	miR-421	PDCD4	Inhibit cell proliferation and induce cell apoptosis	[45]

**Table 2 biomolecules-12-00359-t002:** Target and role of circ-ITCH in cell differentiation.

Disease	miRNA	Related Genes/Pathways	Function Role	Reference
Osteoporosis	miR-214	YAP1	Promote the osteogenic differentiation	[47]
/	miR-34a and miR-146a	MAPK pathway	Regulate PDLSC osteogenic differentiation	[46]

**Table 3 biomolecules-12-00359-t003:** Target and role of circ-ITCH in inflammation.

Disease	miRNA	Related Genes	Function Role	Reference
Acute lung injury	miR-382-5p	TOP1	Alleviate ALI	[38]
Intervertebral disc degeneration	/	TNF-α, IL-1β	Ameliorate IVDD	[44]
Diabetic nephropathy	miR-33a-5p	SIRT6	Alleviate renal inflammation and fibrosis	[51]
Diabetic retinopathy	miR-22	MMP-2, MMP-9, TNF-α	Prevent neovascularization and inflammation	[55]

**Table 4 biomolecules-12-00359-t004:** Target and role of circ-ITCH in drug resistance/toxicity.

Disease	miRNA	Related Genes	Function Role	Reference
Osteosarcoma	miR-524	RASSF6	Promote doxorubicin sensitivity	[56]
Doxorubicin-induced cardiotoxicity	miR-330-5p	SIRT6, Survivin, SERCA2a	Ameliorate Doxorubicin-induced cardiotoxicity	[57]
Multiple myeloma	miR-615-3p	PRKCD	Increase bortezomib sensitivity	[58]
Castration-resistant prostate cancer	miR-17	/	Inhibit the proliferation, migration, and invasion	[59]

**Table 5 biomolecules-12-00359-t005:** Target and role of circ-ITCH in tumorigenesis.

Invasion and Metastasis Regulation				
Disease	miRNA	Related Genes	Function Role	Reference
Prostate cancer	/	/	Correlate with low lymph mode metastasis risk	[60]
Gastric cancer	miR-199a-5p	Klotho	Suppress metastasis	[61]
Osteosarcoma	miR-7	EGFR	Promote migration and invasion	[62]
Cervical cancer	miR-93-5p	FOXK2	Suppress proliferation, migration, and invasion	[63]
Oral squamous cell carcinoma	miR-421	PDCD4	Inhibit cell proliferation and induce cell apoptosis	[64]
**Cell Metabolism Regulation**				
**Disease**	**miRNA**	**Related Genes**	**Function Role**	**Reference**
Ovarian cancer	miR-106a	CDH1	Suppress proliferation, invasion, and glycolysis	[65]
Melanoma	/	GLUT1	Downregulate and suppress glucose uptake	[66]

## Data Availability

All data included in this study are available upon request by contact with the corresponding author.

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
