# Peer review of "Circular RNA ITCH: An Emerging Multifunctional Regulator"

_biomolecules, 2022, doi:10.3390/biom12030359_

Round 1

Reviewer 1 Report

Suggested editorial changes. Additions are in red and underlined, deletions are crossed out and underlined.:

Page 1

Line 43; RN needs to be RNA

Page 3

Line 93;  …by acting as a miRNA sponges for such as miR-7…..

Line 101; Besides  Also,…..

Lines 107-8; The Wnt signaling pathway is highly conservative conserved in species’ evolution…..

Page 4

Line 117;  As the most the  The classical….

Line 118; (Figure 3), it exerts…

Line 128; ..to affect glioma cells….

Line 129;  Besides, Knockdown….

Line 145; …myocardial cells

Page 5

Line 155; Besides, Downregulation…

Lines 172-74;  replace the sentence with the following: circ-ITCH appears to play a significant role in development and cell differentiation, however, the mechanism is not fully known and requires further study (Table 2).

Page 6

Line 183; .. be a therapeutic potential target….

Line 207; …has been verified demonstrated..

Line 211; ….ITCH was downregulated…

Lines 212-14; In When DXR-induced….cardiomyocytes were introduced into in cardiotoxic mice….

Page 7

Line 2219; …we believe that the potential role of circ-ITCH…

Line 231; Besides,  Also,…

Line 235 Likewise,  Finally,…

Page 8

Lines 276-79; Replace the two sentences with the following: Zhang et al. demonstrated that circ-ITCH could reverse myocardial ischemia-reperfusion injury (I/R) via the miR-17-5p/Wnt/β-catenin pathway in  H2O2-treated H9c2 cells (a rat myocardial cell line), an in vitro model of myocardial I/R (39).     

Lines 283-84; It is not clear how circ-ITCH would be used therapeutically. However, it could be used as a therapeutic target.

Page 9

Line 300; Low levels of circ-ITCH was were expressed…

Line 313 …circ-ITCH is mainly acts….

Line 316; …signaling pathways

Lines 321-22; By preforming deep conducting transcriptome…

Line 328; …low expression of….

Line 331; Besides,  Also,

Page 10

Line 342; Luo et al. utilized studied …

Lines 344-45; Subsequently Sequestering mi R-106a by circ-ITCH to upregulate CDH1 expression circ-                ITCH inhibits….

Lines 349-50; By Transcriptome….experiments indicate that circ-ITCH….

Lines 358-60; Both the q-PCR results of Chen et al.’s with 48 glioma patients (37) and Li et al.’s primary glioma tissues (86) ………(37,86).

The Perspectives and Conclusions section is poorly written. Lines 380-390; the English is poor and the concepts unclear. The first paragraph of the section makes it sound like the review is only about solid tumors. The section needs a total reorganization and a rewrite.

Suggested editorial changes. Additions are in red and underlined, deletions are crossed out and underlined.:

Page 1

Line 43; RN needs to be RNA

Page 3

Line 93;  …by acting as a miRNA sponges for such as miR-7…..

Line 101; Besides  Also,…..

Lines 107-8; The Wnt signaling pathway is highly conservative conserved in species’ evolution…..

Page 4

Line 117;  As the most the  The classical….

Line 118; (Figure 3), it exerts…

Line 128; ..to affect glioma cells….

Line 129;  Besides, Knockdown….

Line 145; …myocardial cells

Page 5

Line 155; Besides, Downregulation…

Lines 172-74;  replace the sentence with the following: circ-ITCH appears to play a significant role in development and cell differentiation, however, the mechanism is not fully known and requires further study (Table 2).

Page 6

Line 183; .. be a therapeutic potential target….

Line 207; …has been verified demonstrated..

Line 211; ….ITCH was downregulated…

Lines 212-14; In When DXR-induced….cardiomyocytes were introduced into in cardiotoxic mice….

Page 7

Line 2219; …we believe that the potential role of circ-ITCH…

Line 231; Besides,  Also,…

Line 235 Likewise,  Finally,…

Page 8

Lines 276-79; Replace the two sentences with the following: Zhang et al. demonstrated that circ-ITCH could reverse myocardial ischemia-reperfusion injury (I/R) via the miR-17-5p/Wnt/β-catenin pathway in  H2O2-treated H9c2 cells (a rat myocardial cell line), an in vitro model of myocardial I/R (39).     

Lines 283-84; It is not clear how circ-ITCH would be used therapeutically. However, it could be used as a therapeutic target.

Page 9

Line 300; Low levels of circ-ITCH was were expressed…

Line 313 …circ-ITCH is mainly acts….

Line 316; …signaling pathways

Lines 321-22; By preforming deep conducting transcriptome…

Line 328; …low expression of….

Line 331; Besides,  Also,

Page 10

Line 342; Luo et al. utilized studied …

Lines 344-45; Subsequently Sequestering mi R-106a by circ-ITCH to upregulate CDH1 expression circ-                ITCH inhibits….

Lines 349-50; By Transcriptome….experiments indicate that circ-ITCH….

Lines 358-60; Both the q-PCR results of Chen et al.’s with 48 glioma patients (37) and Li et al.’s primary glioma tissues (86) ………(37,86).

The Perspectives and Conclusions section is poorly written. Lines 380-390; the English is poor and the concepts unclear. The first paragraph of the section makes it sound like the review is only about solid tumors. The section needs a total reorganization and a rewrite.

Suggested editorial changes. Additions are in red and underlined, deletions are crossed out and underlined.:

Page 1

Line 43; RN needs to be RNA

Page 3

Line 93;  …by acting as a miRNA sponges for such as miR-7…..

Line 101; Besides  Also,…..

Lines 107-8; The Wnt signaling pathway is highly conservative conserved in species’ evolution…..

Page 4

Line 117;  As the most the  The classical….

Line 118; (Figure 3), it exerts…

Line 128; ..to affect glioma cells….

Line 129;  Besides, Knockdown….

Line 145; …myocardial cells

Page 5

Line 155; Besides, Downregulation…

Lines 172-74;  replace the sentence with the following: circ-ITCH appears to play a significant role in development and cell differentiation, however, the mechanism is not fully known and requires further study (Table 2).

Page 6

Line 183; .. be a therapeutic potential target….

Line 207; …has been verified demonstrated..

Line 211; ….ITCH was downregulated…

Lines 212-14; In When DXR-induced….cardiomyocytes were introduced into in cardiotoxic mice….

Page 7

Line 2219; …we believe that the potential role of circ-ITCH…

Line 231; Besides,  Also,…

Line 235 Likewise,  Finally,…

Page 8

Lines 276-79; Replace the two sentences with the following: Zhang et al. demonstrated that circ-ITCH could reverse myocardial ischemia-reperfusion injury (I/R) via the miR-17-5p/Wnt/β-catenin pathway in  H2O2-treated H9c2 cells (a rat myocardial cell line), an in vitro model of myocardial I/R (39).     

Lines 283-84; It is not clear how circ-ITCH would be used therapeutically. However, it could be used as a therapeutic target.

Page 9

Line 300; Low levels of circ-ITCH was were expressed…

Line 313 …circ-ITCH is mainly acts….

Line 316; …signaling pathways

Lines 321-22; By preforming deep conducting transcriptome…

Line 328; …low expression of….

Line 331; Besides,  Also,

Page 10

Line 342; Luo et al. utilized studied …

Lines 344-45; Subsequently Sequestering mi R-106a by circ-ITCH to upregulate CDH1 expression circ-                ITCH inhibits….

Lines 349-50; By Transcriptome….experiments indicate that circ-ITCH….

Lines 358-60; Both the q-PCR results of Chen et al.’s with 48 glioma patients (37) and Li et al.’s primary glioma tissues (86) ………(37,86).

The Perspectives and Conclusions section is poorly written. Lines 380-390; the English is poor and the concepts unclear. The first paragraph of the section makes it sound like the review is only about solid tumors. The section needs a total reorganization and a rewrite.

Suggested editorial changes. Additions are in red and underlined, deletions are crossed out and underlined.:

Page 1

Line 43; RN needs to be RNA

Page 3

Line 93;  …by acting as a miRNA sponges for such as miR-7…..

Line 101; Besides  Also,…..

Lines 107-8; The Wnt signaling pathway is highly conservative conserved in species’ evolution…..

Page 4

Line 117;  As the most the  The classical….

Line 118; (Figure 3), it exerts…

Line 128; ..to affect glioma cells….

Line 129;  Besides, Knockdown….

Line 145; …myocardial cells

Page 5

Line 155; Besides, Downregulation…

Lines 172-74;  replace the sentence with the following: circ-ITCH appears to play a significant role in development and cell differentiation, however, the mechanism is not fully known and requires further study (Table 2).

Page 6

Line 183; .. be a therapeutic potential target….

Line 207; …has been verified demonstrated..

Line 211; ….ITCH was downregulated…

Lines 212-14; In When DXR-induced….cardiomyocytes were introduced into in cardiotoxic mice….

Page 7

Line 2219; …we believe that the potential role of circ-ITCH…

Line 231; Besides,  Also,…

Line 235 Likewise,  Finally,…

Page 8

Lines 276-79; Replace the two sentences with the following: Zhang et al. demonstrated that circ-ITCH could reverse myocardial ischemia-reperfusion injury (I/R) via the miR-17-5p/Wnt/β-catenin pathway in  H2O2-treated H9c2 cells (a rat myocardial cell line), an in vitro model of myocardial I/R (39).     

Lines 283-84; It is not clear how circ-ITCH would be used therapeutically. However, it could be used as a therapeutic target.

Page 9

Line 300; Low levels of circ-ITCH was were expressed…

Line 313 …circ-ITCH is mainly acts….

Line 316; …signaling pathways

Lines 321-22; By preforming deep conducting transcriptome…

Line 328; …low expression of….

Line 331; Besides,  Also,

Page 10

Line 342; Luo et al. utilized studied …

Lines 344-45; Subsequently Sequestering mi R-106a by circ-ITCH to upregulate CDH1 expression circ-                ITCH inhibits….

Lines 349-50; By Transcriptome….experiments indicate that circ-ITCH….

Lines 358-60; Both the q-PCR results of Chen et al.’s with 48 glioma patients (37) and Li et al.’s primary glioma tissues (86) ………(37,86).

The Perspectives and Conclusions section is poorly written. Lines 380-390; the English is poor and the concepts unclear. The first paragraph of the section makes it sound like the review is only about solid tumors. The section needs a total reorganization and a rewrite.

Author Response

Cover Letter

Manuscript ID: biomolecules-1606867

Manuscript title: Circular RNA ITCH: An Emerging Multifunctional Regulator

We appreciate the reviewer’s careful reading of our manuscript and thoughtful comments. As indicated below, our manuscript has been revised in response to the editor’s and reviewer’s comments. The Perspectives and Conclusions section has been totally reorganized and rewritten. Further, the manuscript was revised by a native English speaker to improve the clarity of the scientific writing. We mark up all revisions by using the “Track Changes” in the manuscript.

We hope the revised manuscript is acceptable for publication. We are looking forward to your response.

Thank you for your kind consideration of this manuscript.

Best Regards

Reviewer 2 Report

Comment 1.

The revised manuscript may be eligible to be published in the Journal.

Author Response

Cover Letter

Manuscript ID: biomolecules-1606867

Manuscript title: Circular RNA ITCH: An Emerging Multifunctional Regulator

We appreciate the reviewer’s careful reading of our manuscript and thoughtful comments. As indicated below, our manuscript has been revised in response to the editor’s and reviewer’s comments. The Perspectives and Conclusions section has been totally reorganized and rewritten. Further, the manuscript was revised by a native English speaker to improve the clarity of the scientific writing. We mark up all revisions by using the “Track Changes” in the manuscript.

We hope the revised manuscript is acceptable for publication. We are looking forward to your response.

Thank you for your kind consideration of this manuscript.

Best Regards

This manuscript is a resubmission of an earlier submission. The following is a list of the peer review reports and author responses from that submission.

Round 1

Reviewer 1 Report

This manuscript deals with a significant issue. Unfortunately, the authors' ability to present their ideas in proper English is not adequate for publication. Quite honestly, in sections the English is too poor to know what the authors are trying to convey. If they are interested in getting this paper published in an English written journal they must get serious help in editing the paper.

Reviewer 2 Report

Circular RNA ITCH: An Emerging Multifunctional Regulator

Authors in this review have elaborated the role of circular RNA itchy E3 ubiquitin protein ligase in different cellular and disease condition. It is interesting to note that Cir-ITCH is responsible to regulate cancer cell metastasis, invasion, proliferation, apoptosis, and metabolism by sponging different miRNAs. It even plays a role in cardiovascular and bone diseases. Although the review is well written and merit publication, below I will highlight some of the drawbacks which need to be rectified before the manuscript can be considered for publication.

  1. There are major English grammatical errors which makes it hard to understand what the authors are trying to convey. So, I would request the authors to seek help of an English language expert to rectify this mistake.
  2. Line 105 – authors claim ITCH protein degrades phosphorylated dishevelled thereby stabilizing beta-catenin protein in free state. This statement can be further from truth as degradation of dishevelled inhibits APC/Axin/GSK3β proteolytic complex which degrades beta-catenin. Thus, ITCH mediated degradation of dishevelled should lead to beta-catenin degradation not stabilization.
  3. Figure 2 – The message conveyed by this figure suggest the same thing I elaborated in my point above. It shows Dvl promoting APC/Axin/GSK3β proteolytic complex. Rather I would request the authors to change it to show Dvl inhibiting APC/Axin/GSK3β proteolytic complex leading to beta-catenin free state stabilization which is how canonical Wnt pathway works.
  4. Line 148, 149 – Bax, cleaved caspase 3 and cleaved caspase 9 are termed as anti-apoptotic protein whereas they are apoptotic proteins. So, this mistake needs to be corrected.
  5. Line 203 - Authors should use shorthand after first introducing the term such as for competing endogenous RNA (ceRNA).

Reviewer 3 Report

Comment 1.

The authors should add a Figure explaining a mechanism underlying the generation of the circular RNA ITCH on the human genome.

Comment 2. 

There are many English grammatical errors in the manuscript. Authors should edit the manuscript according to standard scientific English grammar.